# Hyperspectral Imaging for Assessment of Initial Graft Function in Human Kidney Transplantation

**DOI:** 10.3390/diagnostics12051194

**Published:** 2022-05-10

**Authors:** Sophie Romann, Tristan Wagner, Shadi Katou, Stefan Reuter, Thomas Vogel, Felix Becker, Haluk Morgul, Philipp Houben, Philip Wahl, Andreas Pascher, Sonia Radunz

**Affiliations:** 1Department of General, Visceral and Transplant Surgery, University Hospital Münster, 48149 Münster, Germany; sophie.romann@uni-muenster.de (S.R.); tristan.wagner@ukmuenster.de (T.W.); shadi.katou@ukmuenster.de (S.K.); thomas.vogel@ukmuenster.de (T.V.); felix.becker@ukmuenster.de (F.B.); haluk.morguel@ukmuenster.de (H.M.); philipp.houben@ukmuenster.de (P.H.); andreas.pascher@ukmuenster.de (A.P.); 2Department of General Internal Medicine, Nephrology and Rheumatology, University Hospital Münster, 48149 Münster, Germany; stefan.reuter@ukmuenster.de; 3Diaspective Vision GmbH, 18233 Am Salzhaff, Germany; philip.wahl@diaspective-vision.com

**Keywords:** hyperspectral imaging, delayed graft function, hypothermic machine perfusion, kidney transplantation, organ preservation

## Abstract

The aim of our study was to evaluate hyperspectral imaging (HSI) as a rapid, non-ionizing technique for the assessment of organ quality and the prediction of delayed graft function (DGF) in kidney transplantation after static cold storage (SCS, n = 20), as well as hypothermic machine perfusion (HMP, n = 18). HSI assessment of the kidney parenchyma was performed during organ preservation and at 10 and 30 min after reperfusion using the TIVITA^®^ Tissue System (Diaspective Vision GmbH, Am Salzhaff, Germany), calculating oxygen saturation (StO_2_), near-infrared perfusion index (NIR), tissue haemoglobin index (THI), and tissue water index (TWI). Recipient and donor characteristics were comparable between organ preservation groups. Cold ischemic time was significantly longer in the HMP group (14.1 h [3.6–23.1] vs. 8.7h [2.2–17.0], *p* = 0.002). The overall presence of DGF was comparable between groups (HMP group n = 10 (55.6%), SCS group n = 10 (50.0%)). Prediction of DGF was possible in SCS and HMP kidneys; StO_2_ at 10 (50.00 [17.75–76.25] vs. 63.17 [27.00–77.75]%, *p* = 0.0467) and 30 min (57.63 [18.25–78.25] vs. 65.38 [21.25–83.33]%, *p* = 0.0323) after reperfusion, as well as NIR at 10 (41.75 [1.0–58.00] vs. 48.63 [12.25–69.50], *p* = 0.0137) and 30 min (49.63 [8.50–66.75] vs. 55.80 [14.75–73.25], *p* = 0.0261) after reperfusion were significantly lower in DGF kidneys, independent of the organ preservation method. In conclusion, HSI is a reliable method for intraoperative assessment of renal microperfusion, applicable after organ preservation through SCS and HMP, and predicts the development of DGF.

## 1. Introduction

Hyperspectral imaging (HSI) demonstrated promising results for the characterization of tissues and the assessment of physiologic tissue parameters [1,2]. In contrast to other intraoperative imaging methods, HSI is contactless, non-invasive, non-ionizing, and the administration of a contrast medium is not required [3]. The measurements are taken within a few seconds; therefore, the surgical procedure is only marginally disturbed [1].

HSI is capable of providing quantitative diagnostic information on tissue composition, morphology, and pathology. Analysis software provides an RGB image and four false-color images representing physiologic parameters of the recorded tissue area; these parameters are oxygen saturation (StO_2_), near-infrared perfusion index (NIR), tissue hemoglobin index (THI), and tissue water index (TWI) [4]. StO2 (%) represents the relative blood oxygenation in the microcirculation of superficial layers (approximately 1 mm), while the NIR perfusion index (0–100) analyzes tissue layers at 4–6 mm penetration depth. The indices THI (0–100) and TWI (0–100) display the distribution of hemoglobin and water in the observed tissue area, respectively [5].

Ischemia-reperfusion injury in kidney transplantation is known to be a key factor in the development of delayed graft function (DGF). DGF—which develops in 16–29% of kidney transplants [6,7]—is associated with an inferior one-year graft function, as well as a poorer overall graft and patient survival [8,9]. Kidney allograft evaluation for prediction of DGF is usually based on donor characteristics and histological assessment of zero-time biopsies [10]. Currently, there is no universal non-contact method for reliable prediction of DGF.

Tools for objective intraoperative assessment of graft viability and performance are lacking in kidney transplantation. HSI was proven as an intraoperative real-time assessment tool delivering quantitative information on organ viability and performance [2,11,12,13,14]. In a pilot study, HSI was applied in 17 kidney grafts, 2 of which had DGF; after reperfusion, the DGF kidneys displayed significantly decreased allograft oxygenation and microperfusion [2].

Hence, the objective of our study was to evaluate HSI in real-life kidney transplantation for the assessment of organ quality and the prediction of DGF. The secondary objective was to evaluate its applicability in different organ preservation methods.

## 2. Materials and Methods

Patients who underwent deceased-donor kidney transplantation at our center between March 2021 and December 2021 were eligible for the study. Recipients were at least 18 years old, and were wait-listed for kidney transplantation with Eurotransplant. The study was conducted in accordance with the declaration of Helsinki, and was approved by the local ethics committee (ID 2021-223-f-S). The requirement for informed consent for the study was waived.

Following organ procurement, kidney allografts were preserved through static cold storage (SCS), using histidine-tryptophan-ketoglutarate solution (Custodiol^®^ HTK Solution) or University of Wisconsin solution (Belzer UW^®^ Cold Storage Solution) for organ preservation during transportation from the donor to the recipient hospital. Further organ preservation was performed by means of SCS or hypothermic machine perfusion (HMP), depending on recipient factors, e.g., the need for pre-operative dialysis or plasmapheresis, or logistical reasons, e.g., operating room capacity. Kidneys in the HMP group were connected to the Lifeport^®^ Kidney Transporter (Organ Recovery Systems, Chicago, IL, USA) and then perfused at 2–4 °C using one liter of KPS-1 (Organ Recovery Systems, Chicago, IL, USA).

Kidneys were transplanted in the iliac fossa with vascular anastomoses to the external iliac vessels. Ureteroneocystostomy was performed according to the modified Lich-Gregoir technique with insertion of a double-J stent. For immunosuppressive induction, all patients received Thymoglobulin^®^ 1.5 mg/kg on the day of transplantation. Further administration on postoperative day (POD) 1–3 was dependent on the patient’s cellular immune status. Maintenance immunosuppression consisted of tacrolimus (trough level 6–8 ng/mL until month 3, then 5–7 ng/mL), mycophenolate-mofetil, and steroids.

Hyperspectral images of the kidney parenchyma were acquired after back-table preparation and at 10 and 30 min after reperfusion using the TIVITA^®^ Tissue System (Diaspective Vision GmbH, Am Salzhaff, Germany). For HSI measurement, all surrounding lights were switched off to ensure undisturbed data acquisition. A distance of 50 cm between the kidney allograft and the camera with a focal length of 25 mm was utilized for kidney parenchyma assessment. A camera-specific software (TIVITA^®^ Suite, Diaspective Vision GmbH, Am Salzhaff, Germany) calculated StO_2_, THI, NIR, and TWI [15]. Regions of interest (ROI) were placed equally in the upper and lower kidney pole.

The following recipient characteristics were evaluated: age, sex, body mass index (BMI), kidney disease, and waiting time to kidney transplantation (i.e., dialysis vintage). The following donor variables were assessed: age, sex, height, BMI, history of hypertension, history of diabetes mellitus, CMV status, HCV status, cause of brain death, and serum creatinine levels at procurement. The kidney donor profile index (KDPI) was calculated using the Organ Procurement and Transplantation Network (OPTN) online tool [16]. The KDPI reference population were all kidney donors recovered through OPTN in 2020. The following procurement data and surgical details were collected: method of organ preservation, duration of HMP, cold ischemia time (CIT), warm ischemia time (WIT), and duration of surgery. The following outcome variables were recorded: serum creatinine levels, urea levels and estimated GFR on POD 1, 3 and on the day of discharge, presence of DGF, length of hospital stay, and patient and graft survival. DGF was defined as the need for at least one hemodialysis session during the first week posttransplant [17].

All data were tested for normality using the Shapiro–Wilk normality test. Categorical variables are presented as percentages and continuous variables as median [range], unless stated otherwise. Differences were tested using Student’s *t*-test or Mann–Whitney-U tests, as appropriate. Graft and patient survival were evaluated using the Kaplan–Meier method, and compared with the log-rank test. The reference point for all calculations of survival was the day of kidney transplantation. Overall graft survival was determined until death, return to dialysis, or the end of the study period. A *p* value of ≤0.05 (two-tailed) was considered significant.

Data collection and statistical analysis were performed using Microsoft Excel 2010 (Microsoft Corporation, Redmond, WA, USA), IBM SPSS Statistics (version 23.0 for Windows, SPSS, Inc., Chicago, IL, USA), and GraphPad Prism 9 for macOS version 9.3.1 (GraphPad Software, San Diego, CA, USA).

## 3. Results

We included 38 recipients of deceased-donor kidney allografts in our study. All donor organs were derived from donation after brain death. The most frequent kidney diseases were glomerulonephritis (n = 17, 44.8%) and vascular nephropathy (n = 8, 21.1%). The median dialysis vintage was 67 [16–181] months. The method of organ preservation was SCS in 20 (52.6%) cases and HMP in 18 (47.4%) cases. Recipient and donor characteristics, as well as surgical details, are given in Table 1. Both groups were comparable, with the exception of CIT being significantly longer in the HMP Group (14.1 [3.6–23.1] vs. 8.7 [2.2–17.0], *p* = 0.0002).

A total of 20 patients (50.6%) developed DGF (SCS Group n = 10 (50.0%), HMP Group n = 10 (55.7%), *p* = 0.7568). Kidney allografts developing DGF were more likely to be from male donors. Furthermore, there was also a trend towards a higher recipient BMI and for recipients being male more frequently; however, this did not reach statistical significance (Table 2). KDPI was comparable between groups. Upon HSI assessment, kidney allografts with DGF displayed significantly lower StO_2_ and NIR perfusion indices at 10 and 30 min after reperfusion and a significant lower TWI at 30 min after reperfusion, independent of the organ preservation method (Table 3, Figure 1).

HMP-preserved kidneys showed increased StO_2_ levels compared to SCS-preserved kidneys at back-table preparation, i.e., prior to dynamic organ preservation (Table 4). Otherwise, baseline values were comparable between the different groups of organ preservation. After reperfusion, no differences in HSI parameters were detected between the SCS- and HMP-preserved kidney allografts.

The quality of the organ is already evident in the false color-coded hyperspectral image without setting ROI. Figure 2 shows the oxygen saturation of a well perfused, primary functioning kidney 30 min after reperfusion. The organ displays an evenly distributed red color equal to an oxygen saturation of 80–100%. In comparison, Figure 3 shows a DGF kidney with low oxygen saturation. The tissue is displayed in green and blue colors, demonstrating an oxygen saturation of 0–40%. Thus, HSI allows the surgeon to obtain a first impression of the organ quality within seconds.

The follow-up of the entire study cohort was 8.8 [3.5–12.7] months. The length of hospital stay was 16 [4–42] days. Creatinine at discharge was significantly higher in the DGF group (3.01 [1.49–9.29] vs. 1.88 [0.70–3.92] mg/dL, *p* = 0.0003). During the study period, no graft loss occurred. All patients are alive and do not need of dialysis treatment.

## 4. Discussion

Our study demonstrates that HSI assessment of human kidney allografts is feasible and reliable, regardless of the organ preservation method. As an easy, non-invasive, non-ionizing technique with a rapid application, HSI does not disturb the surgical procedure. In our study, we demonstrated that HSI can be utilized not only to predict DGF in SCS-preserved kidneys but also in HMP-preserved kidneys. Despite significantly longer CIT, HMP kidneys yielded comparable results on HSI assessment in the recipient, confirming the attributed positive effects of HMP on microperfusion [18].

DGF not only affects the kidney allograft and the recipient in the immediate post-transplant period, but is also a predictor for the allograft’s subsequent course, which is frequently characterized by long-term detrimental effects, e.g., recurrent episodes of acute rejection, declining kidney function, and impaired graft survival [9]. Despite the associated increased risk for DGF, non-standard criteria donor organs have become the current norm. This highlights the need for timely diagnosis and treatment of DGF to ensure long-term kidney graft survival.

Kidney allografts with measurably impaired microperfusion are prone to developing DGF [19]. However, reliable tools for the direct assessment of kidney allograft microper-fusion are not yet in place. The common way to obtain information about graft viability is still to interpret clinical donor parameters or to perform zero-time biopsies [20]. HSI showed promising first results as a potential tool to measure the microperfusion of SCS-preserved kidneys [2].

Our results confirm a significantly decreased microperfusion in DGF kidneys after reperfusion in the recipient, as assessed by StO_2_ and NIR 10 and 30 min after reperfusion. Furthermore, a significantly decreased TWI was evident in DGF kidneys 30 min after reperfusion. In kidneys developing DGF, vasoconstriction and renal edema may inhibit the capillary flow after reperfusion in the recipient, resulting in decreased oxygen and nutrient delivery, as evident in decreased NIR and StO_2_ values in HSI measurements. In primary function kidney grafts, vasodilation probably not only improves nutrient supply, but may also lead to an increased water content, as evident in increased TWI values. Since the higher values for TWI go along with higher values in StO_2_ and NIR, the increased water content is judged as a positive factor, and not as a sign for renal edema. For this difference in water content to become apparent in HSI measurements, a stable period of adequate perfusion appears to be necessary; therefore, we recommend additional HSI measurements 30 min after reperfusion.

Altogether, HSI offers valuable information on allograft microperfusion. Even in kidney allografts with a supposedly enhanced microperfusion after undergoing dynamic organ preservation, i.e., HMP, HSI detects significant differences in allografts with and without DGF. Nevertheless, there are some drawbacks to HSI assessment, as one needs to obtain a close, direct view of the organ, and transcutaneous measurements are not possible. Pictures are static, and a continuous sequence of moving images is not possible.

Our study is limited to 38 kidneys without randomization, and was performed at a single center, which may introduce bias. Nevertheless, this is the largest study cohort analyzing HSI in human kidney transplantation after SCS and HMP. Due to the high DGF rate in our study, regardless of the preservation method, differences in HSI measurements according to initial graft function are clearly detectable despite the small sample size. HSI measurements at back-table preparation detecting differences in tissue water content or hemoglobin deposit might be promising for the prediction of DGF prior to transplantation, but this needs to be verified in larger samples. HSI assessment might also be applicable during oxygenated HMP for detecting kidney allografts at increased risk for developing DGF.

A further possible field for HSI application might be the evaluation of the organ quality prior to procurement from the deceased donor. In this setting, the HSI technique might save resources by preventing the transplantation of organs of truly poor quality that are also not amenable to organ reconditioning. However, the surgical process during the procurement procedure would need to be adapted for the HSI measurements, with early removal of the kidneys’ fat capsule for direct view of the parenchyma.

## 5. Conclusions

HSI is a reliable and simple tool for gathering information on the microperfusion of transplanted kidneys, regardless of the preservation method, and therefore adds substantial information to routinely performed sonography. Since the quality of the microperfusion is crucial for the detection of ischemia and reperfusion injury, and also predicts initial graft performance, HSI is a valuable addition to established organ quality prediction techniques, such as KDPI and zero-time biopsies.

## Figures and Tables

**Figure 1 diagnostics-12-01194-f001:**
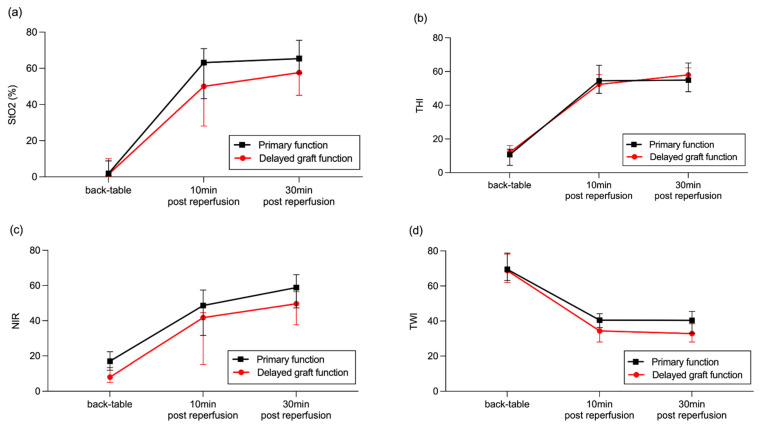
HSI assessment according to initial graft function: (**a**) StO_2_, (**b**) THI, (**c**) NIR, (**d**) TWI. Data are presented as median (interquartile range).

**Figure 2 diagnostics-12-01194-f002:**
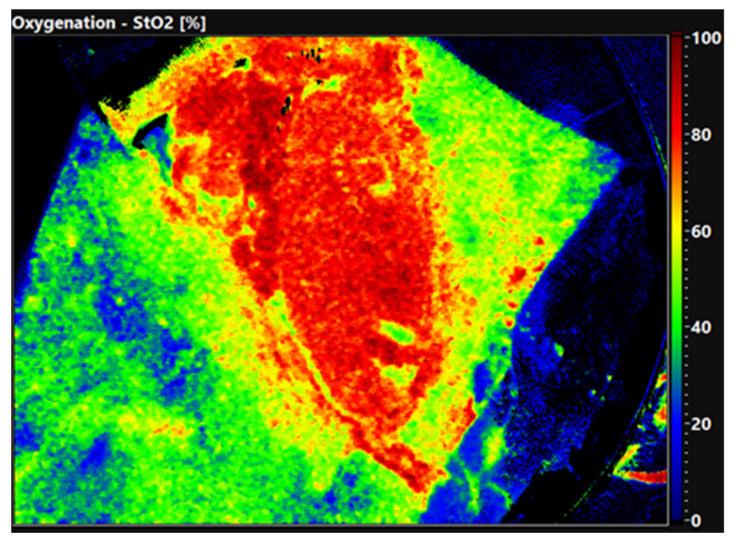
False-color image of a primary functioning kidney 30 min after reperfusion. The red color displays a high oxygen saturation (80–100%).

**Figure 3 diagnostics-12-01194-f003:**
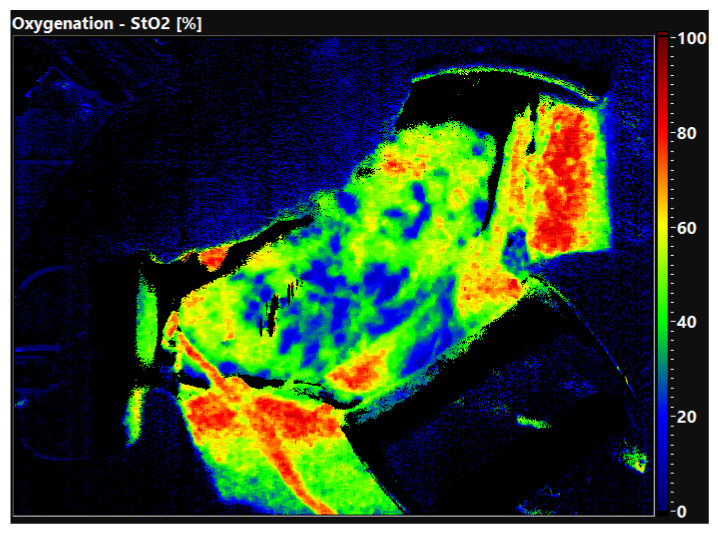
False-color image of a kidney allograft with DGF 30 min after reperfusion. The blue and green colours display a low oxygen saturation (0–40%).

**Table 1 diagnostics-12-01194-t001:** Recipient and donor characteristics, as well as surgical details, according to organ preservation method.

	SCS (n = 20)	HMP (n = 18)	*p*
Recipient age (yrs)	62 [25–74]	57 [27–72]	0.4902
Recipient male sex (%)	60.0	50.0	0.5359
Recipient BMI (kg/m^2^)	24.7 [17.1–31.6]	25.4 [18.4–33.1]	0.7207
KDPI (%)	76 [15–99]	68 [6–99]	0.4632
CIT (h)	7.2 [2.2–17.0]	15.2 [6.6–23.1]	**0.0002**
HMP (h)	n.a.	8.7 [2.9–15.3]	
WIT (min)	40 [22–59]	35 [22–50]	0.0995
DGF (%)	50.0	55.6	0.7568

(SCS, static cold storage; HMP, hypothermic machine perfusion; BMI, body mass index; KDPI, kidney donor profile index; CIT, cold ischemia time; WIT, warm ischemia time; DGF, delayed graft function).

**Table 2 diagnostics-12-01194-t002:** Recipient and donor characteristics, as well as surgical details, according to initial graft function.

	Primary Function (n = 18)	DGF (n = 20)	*p*
Recipient age (yrs)	63 [25–74]	57 [27–72]	0.9739
Recipient male sex (%)	38.9	70.0	0.0541
Recipient BMI (kg/m2)	24.2 [17.1–31.1]	27.3 [18.6–33.1]	0.0593
Donor male sex (%)	33.3	70.0	**0.0496**
KDPI (%)	75 [6–99]	72 [21–98]	0.7729
CIT (h)	9.8 [4.5–17.2]	12.0 [2.2–23.1]	0.9942
HMP (h)	9.1 [4.0–15.3]	8.2 [2.9–14.9]	0.7890
WIT (min)	36 [22–48]	39 [23–59]	0.3495

(DGF, delayed graft function; BMI, body mass index; KDPI, kidney donor profile index; CIT, cold ischemia time; WIT, warm ischemia time; HMP, hypothermic machine perfusion).

**Table 3 diagnostics-12-01194-t003:** HSI parameters stratified according to initial graft function.

	Primary Function (n = 18)	DGF (n = 20)	*p*
After back-table preparation			
StO_2_ (%)	1.88 [0.20–36.75]	1.38 [0.00–28.60]	0.6176
THI	10.67 [1.25–26.19]	12.13 [4.25–22.00]	0.2050
NIR	17.00 [1.00–38.75]	8.01 [0.00–25.75]	**0.0068**
TWI	69.50 [59.80–92.00]	68.78 [57.75–88.50]	0.7386
10 min after reperfusion			
StO_2_ (%)	63.17 [27.00–77.75]	50.00 [17.75–76.25]	**0.0467**
THI	54.50 [29.80–83.00]	52.38 [33.25–65.00]	0.3130
NIR	48.63 [12.25–69.50]	41.75 [1.0–58.00]	**0.0137**
TWI	40.50 [15.00–55.50]	34.38 [15.00–55.00]	0.1433
30 min after reperfusion			
StO_2_ (%)	65.38 [21.25–83.33]	57.63 [18.25–78.25]	**0.0323**
THI	54.88 [16.2–91.00]	58.00 [39.2–70.25]	0.7074
NIR	55.80 [14.75–73.25]	49.63 [8.50–66.75]	**0.0261**
TWI	40.38 [13.25- 61.50]	32.83 [17.75–41.75]	**0.0338**

(DGF, delayed graft function; StO_2_, oxygen saturation; NIR, near-infrared perfusion index; THI, tissue haemoglobin index; TWI, tissue water index).

**Table 4 diagnostics-12-01194-t004:** HSI parameters stratified according to organ preservation method.

	SCS (n = 20)	HMP (n = 18)	*p*
After back-table preparation			
StO_2_ (%)	0.84 [0.00–27.50]	7.00 [0.00–36.75]	**0.0275**
THI	10.67 [1.25–22.00]	11.75 [2.25–26.19]	0.2361
NIR	14.35 [1.00–38.75]	10.88 [0.00–25.75]	0.6694
TWI	69.75 [59.50–92.00]	68.90 [57.75–85.75]	0.9379
10 min after reperfusion			
StO_2_ (%)	58.13 [17.75–77.75]	50.75 [25.50–76.25]	0.4914
THI	52.50 [29.80–83.00]	56.00 [33.25–80.50]	0.6387
NIR	43.88 [1.00–69.50]	40.95 [8.75–62.25]	0.4440
TWI	36.59 [15.00–48.33]	38.88 [20.00–55.50]	0.3145
30 min after reperfusion			
StO_2_ (%)	62.00 [18.25–83.33]	60.63 [27.25–79.00]	0.9137
THI	57.25 [16.20–91.00]	58.08 [39.75–82.00]	0.2862
NIR	53.00 [8.50–73.25]	53.75 [12.20–69.50]	0.9195
TWI	35.13 [13.25–47.67]	38.00 [21.00–61.50]	0.4608

(SCS, static cold storage; HMP, hypothermic machine perfusion; StO_2_, oxygen saturation; NIR, near-infrared perfusion index; THI, tissue haemoglobin index; TWI, tissue water index).

## Data Availability

The data that support the findings of this study are available from the corresponding author upon reasonable request.

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
