# Peer review of "Hyperspectral Imaging for Assessment of Initial Graft Function in Human Kidney Transplantation"

_diagnostics, 2022, doi:10.3390/diagnostics12051194_

Round 1

Reviewer 1 Report

  1. DGF has different and heterogenous definitions among studies. Please additionally give definition of your DGF in the method section.
  2. It’s not clear how KDPI was derived without reference in their “data source and study population” section, and without explicitly stating the KDPI reference population that KDRI was mapped to.
  3. Figure 1 will be better with color form
  4. In Table 1, please use capital letter at the start of each 

Author Response

We thank the reviewer for the overall favorable assessment of our work and the important comments made. The remarks were addressed and the information was added in the manuscript as suggested:

  1. The applied definition of DGF is already given in the methods section (p. 3, lines 106-107: "DGF was defined as the need for at least one hemodialysis session during the first week posttransplant [17].").
  2. In the methods section, method of KDPI calculation and the reference for the online calculator tool are already given (p. 3 lines 100-101: "The kidney donor profile index (KDPI) was calculated using the Organ Procurement and Transplantation Network (OPTN) online tool [16]."). The KDPI reference population are all kidney donors recovered through OPTN in 2020, i.e. the reference population for KDRI mapping is a North American population, not a European population.
  3. Figure 1 is now displayed in color.
  4. In Table 1, capital letters are used at the start of each row.

Reviewer 2 Report

The study is an humble and simple observation of Kidney microcirculation aim at evaluating hyperspectral imaging (HSI) as a rapid assessment of organ quality of delayed graft in kidney transplantation.
The authors reported that HSI is a reliable method for intraoperative assessment of renal microperfusion, applicable after organ preservation by SCS and HMP, and predicts the development of DGF.

Overall HSI is a novel approach, thus every clinical observation under this umbrella, are interesting for the community.

No further comments to add

Author Response

We thank the reviewer for the favourable assessment of our work.